# Steady-State Hydraulic Analysis of High-Rise Building Wastewater Drainage Networks: Modelling Basis

**Colin Stewart \*, Michael Gormley** 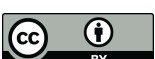**, Yunpeng Xue** ⓘ**, David Kelly** ⓘ **and David Campbell**

Institute for Sustainable Building Design, Heriot-Watt University, Edinburgh EH14 4AS, UK; m.gormley@hw.ac.uk (M.G.); Y.Xue@hw.ac.uk (Y.X.); d.a.kelly@hw.ac.uk (D.K.); d.p.campbell@hw.ac.uk (D.C.)
\* Correspondence: colin.stewart@hw.ac.uk

**Abstract:** A model is presented which allows steady-state pressure profiles in high-rise wastewater drainage networks to be related to intake air flowrates and discharge water flowrates. This model is developed using data taken from academic literature, and is based on experimental observations which suggest that a vertical annular downflow develops over distance such that the pressure gradient in the wet stack may be expressed as the sum of junction components and developed flow components. The model is used to analyse a simplified 'medium rise' primary vented system of height 40 m, hosting two inflow junctions, crossvents and Air Admittance Valves (AAVs). The model illustrates how the air supply configuration affects the airflow rates within the stack and the vents, and how the configuration affects the steady-state hydraulic pressure profile. The model offers the possibility of an alternative approach to the design of high-rise wastewater drainage networks, compared to existing design codes. These codes generally do not explain the role that the air admitted into the network has upon its performance.

**Keywords:** two-phase flow modelling; annular flow; flow development; tall building drainage; air ventilation; pipe networks; wastewater; design codes

## 1. Introduction

Tall-building wastewater drainage systems (WDSs) are comprised of wet stacks and networks of air vents. The function of these air vents is to allow wastewater to exit and induced airflow to enter efficiently and safely. Discharges subject the air within these systems to variable forces which initiate acoustic waves or pressure surges. The intensity of these surges generally depends upon the types of appliance being discharged; durations of discharges; the geometry of the appliance branches; the presence of pressure suppression components and, potentially, the condition of the sewer network. Prolonged surges which may result from the 'heavy' loading of a system (i.e., occurring as a result of extended-duration discharges) will generally result in the development of a dynamic suction pressure profile. Pressure surges may be large enough to delay or reverse sanitary water discharge; to initiate vibrations and cause noise; to deplete appliance trap seals by blow-out or siphonage [1], or to deform or rupture pipework [2].

The depletion of trap seals within the system is a particular concern. Empty trap seals provide a path for contaminated air to spread from the stack into the habitable space of a building. This spread is enhanced if local pressure in the stack is higher than the exterior pressure; under these conditions the stack actively expels contaminated air rather than removing it as intended. This 'positive pressure' has been shown to spread aerosolized pathogens [3], and has been confirmed as the source of a SARS outbreak in Hong Kong [4]. Coronaviruses are capable of surviving in sewerage systems for days to weeks [5], and more recently, the SARS Cov-2 pathogen ('COVID-19') has been found in high concentrations in a WC in a hospital building [6]. There is thus strong anecdotal evidence that the spread of COVID-19 is linked to malfunctioning WDSs; this has recently prompted risk assessments

to have been carried out high-rise buildings [7,8]. These assessments indicate that risk is most substantial in the upper floors of high-rise buildings, and is non-negligible.

Steady progress has been made in recent decades in the identification, qualification and mitigation of pressure surges [9]. However, this progress has largely been restricted to low-rise WDSs. Significant gaps remain in understanding of behaviour in high-rise WDSs [10]. These gaps may be attributed the higher flowrates encountered in high-rise systems, greater network complexity, and the increased probability of interaction between discharges.

Operational problems are best avoided by *careful design*, rather than by mitigation following construction. Building drainage system design codes have been developed with this goal in mind and are invaluable tools for building services engineers [11–14]. However, these design codes generally use 'discharge unit techniques' which apply a maximum water discharge rate (estimated by summing discharge units representing average flow of individual appliances) to recommend a stack size for a preferred configuration [15]. The air supply configuration and the quantity of air entrained in the stack are not explicitly discussed within these design calculations. Hence, the design codes relegate a potentially very important aspect of WDS design, having potentially very significant impact for high-rise structures, to a position of triviality. This carries the risk that design might not be optimised, or, possibly, a novel design solution might be overlooked. It is these limitations which have prompt the development of a novel two-phase flow model which is described in this article.

### 1.1. Steady-State Pressure Profile

An important property of an operational drainage stack is its *steady-state pressure profile*. This is a hypothetical pressure profile which arises if it is assumed that the flows of water being discharged into the stack and associated rates of air intake remain perfectly steady over time. Note that, in the case of a high-rise building, these flows may be distributed (i.e., they may enter the stack at multiple locations). The profile provides a simple and convenient way to *visualize* the effectiveness of a design, since pressure extremes can easily be compared against a nominated design criterion, such as for example the 50 mm $H_2O$ pressure head typically provided by UK appliance water trap seals. This visualization is not possible when design codes are used to select a suitable design.

It should be noted that while a steady-state pressure profile provides useful design guidance, it does not on its own, *guarantee* an effective design. A drainage system will generally also require to be protected using active suppression components, or additional ventilation pipework in order to handle the larges surges which may occur if events cause air intake rates to change rapidly over time.

### 1.2. Modelling Software

The discussion above suggests the significant advantage to be gained through the development of a software tool capable of modelling the *two-phase* (air-water) flow within in WDS networks. Figure 1a schematically illustrates the domain over which such a proposed tool would be required to function. The domain is divided into a *steady flow* region, a *transient flow* region, and an overlapping *quasi-steady* flow region where, broadly speaking, the liquid flowrate changes sufficiently slowly over time as to resemble a steady flow. Examples of discharge events which would lead to these types of flows are suggested by the table to the right of the drawing. The transient flow region occupies a far larger space than the steady flow region, reflecting the greater number of ways in which wastewater can be discharged to produce unsteady flow and the greater likelihood that this unsteady flow can occur. However, it significant that from a practical viewpoint, steady flow is far easier to measure using an instrumented test rig and is also far simpler to model.

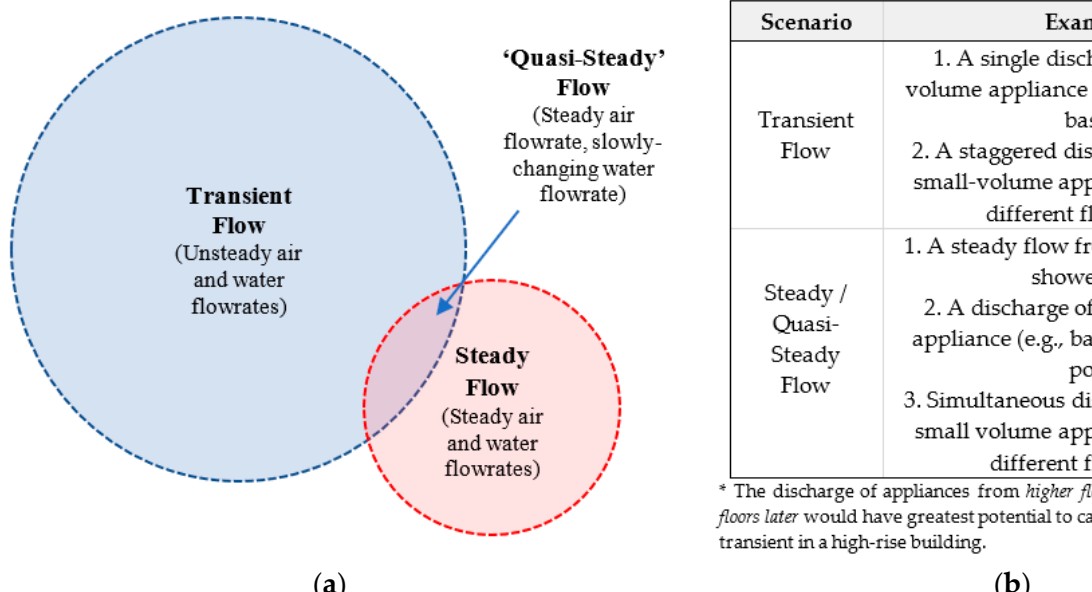

**Figure 1.** Simulation software for design of high-rise WDSs. (**a**) Solver domain (**b**) Examples of discharge events which lie within the solver domain.

These observations motivate the development of a steady-state two-phase hydraulic model, for vertically downwards air-water flow. It is interesting to noted that similar types of tools have been developed for use within the nuclear power and the offshore oil and gas industries [16,17], and have now become integral to design procedures.

## 2. Background

Figure 2 illustrates a flow pattern map for fully developed air-water flow travelling at constant flowrates in vertical-downward pipes at atmospheric pressure [18]. This map illustrates the tendency for fluids to arrange themselves into specific types of flow pattern, or flow geometries, depending upon the normalized flow velocities $U_a$ and $U_w$. These normalized velocities are related to the volumetric flowrates $Q_a$ and $Q_w$ by:

$$U_a = Q_a / A \qquad U_w = Q_w / A \qquad (1)$$

Given sufficiently high values of water velocity $U_w$, Figure 1 indicates that fluids will tend to flow in slug, bubble or churn patterns. In these patterns, the pipe core is either intermittently or permanently blocked by water. However, for lower values of water velocity $U_w$, Figure 1 indicates that fluids will tend to flow in the annular pattern. In the annular flow pattern, liquid is pushed to the wall such that the core remains liquid-free.

Pressure gradients which arise due to bubble, slug and churn flow patterns are typically an *order of magnitude* greater than pressure gradients arising due to an annular flow [19,20]. This observation implies that to minimize the potential for large pressure surges to occur in vertical drainage systems, the stack should be large enough to encourage the annular flow pattern and to discourage the bubble, churn or slug flow patterns as indicated by Figure 1. The orientation of the transition boundary suggests that it easier to satisfy this requirement with a large, normalized air velocity $U_a$; that is to say, if a strong air current can be drawn through the vertical stack. A strong airflow is desirable as it permits a relatively small stack to be used; from an engineering perspective translates into savings in space and material costs.

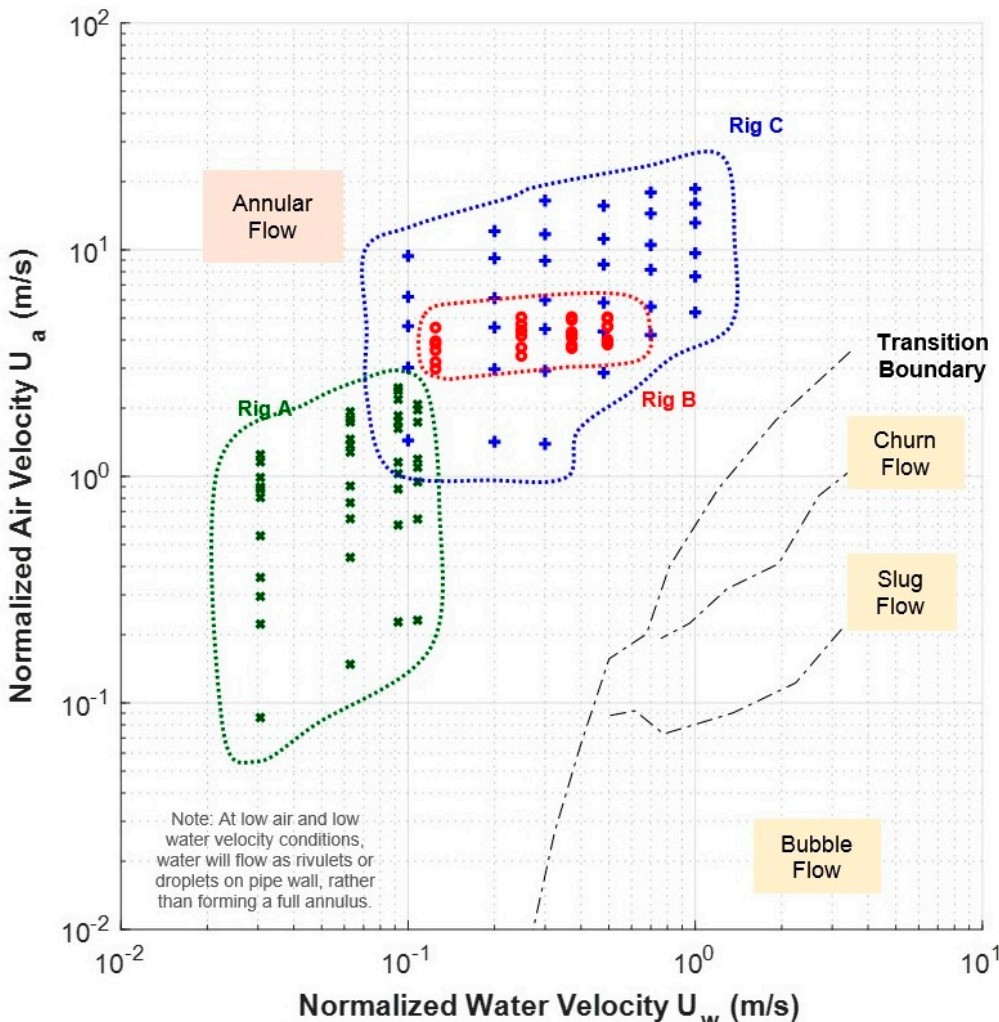

**Figure 2.** Flow pattern map for vertical downward air-water flow [18]. Rig 'A', Rig 'B' and 'C' experimental dataset test conditions shown for reference.

### 2.1. Annular Flow Development

Water commingles with air at injection points, or junctions, in order to create air-water flowing mixtures. The experimental evidence suggests that local flow patterns in the vicinity of such junctions are strongly influenced by the junction geometry, but that these influences diminish with distance travelled downstream. Studies on vertical upwards annular flow suggest that pressure gradient and film volume fraction require around 100–300 diameters to develop from an inlet [21], and that flow asymmetry reduces significantly with distance travelled downstream of U-bends [22]. Studies on vertical downwards annular flow indicate sensitivity of flow development processes to inlet junction geometry [23], and also to flow straightener devices which may be inserted to modify a flow profile [24]. This evidence suggests that an annular flow develops downstream of an inlet, in a manner similar to a single-phase flow, and given a sufficiently long length of pipe, a *fully developed* flow condition will establish. This fully developed flow condition will persist until an interruption such as a bend, a tee section, or a blockage, requires some level of *re-development* to take place.

'Fully developed' annular flow is a quasi-steady condition, in that the phase fractions, phase velocities and pipe pressures are steady in an averaged sense but fluctuate on short timescales. The interface between the liquid annulus and the air core is poorly defined and highly turbulent (as compared to single-phase turbulent flow) and is periodically affected by instabilities such as ripples and roll waves [25,26]. Air in the core travels more rapidly

than the water annulus, and as the boundaries with the churn flow and slug flow regimes are approached, the core absorbs water droplets which travel more rapidly than the air phase [27]. The behaviour of annular flow remains incompletely understood. This means that uncertainty margins in two-phase annular flow models remain large, in comparison to single-phase flow models.

*2.2. Steady-State Hydraulic Model Basis*

Figure 3 shows an element of a developing annular flow within a drainage stack located at a distance $z_j$ downstream from a discharging junction. The water within this element is subject to a body force $dF_w$, a wall shear force $dF_{ws}$ and an interface shear force $dF_i$, while the faces that the bound the element are subject to pressures $P_z$ and $P_{z+\Delta z}$ which are assumed to act uniformly across the element cross-section area. The differences in the face pressures cause net forces to acting upon the air phase, $dF_{P,a}$, and the water phase, $dF_{P,w}$, defined as:

$$dF_{P,a} = -\,\varepsilon\,\Phi A\,dz \qquad dF_{P,w} = -(1-\varepsilon)\,\Phi\,A\,dz \tag{2}$$

where A is the pipe cross-section area (units m$^2$), $\varepsilon$ is the air phase cross-section volume fraction (dimensionless) and where $\Phi$ is the rate of change of pressure with distance $\partial P_z/\partial z$, i.e., the *axial pressure gradient* (units N m$^{-3}$). For steady-state flow conditions, the force balance $dF_w + dF_{ws} + dF_i + dF_{P,w} = 0$ applies.

The *overall* pressure change between a junction located at z = 0 and a point located downstream distance z = $z_j$ may be expressed as the sum of pressure contributions arising due to the junction and due to a developed annular flow. That is to say:

$$P(z_j) = P_j(z_j) + P_d(z_j) \tag{3}$$

where:

$$P(z_j) = \int_{z=0}^{z=z_j} \Phi_j(z)dz + \int_{z=0}^{z=z_j} \Phi_d(z)dz \tag{4}$$

and where $\Phi_j$ and $\Phi_d$ may be described as the *junction component* and the *developed flow component* of the pressure gradient $\Phi$.

Two proposals are now made. Firstly, it is proposed that junction pressure gradient component $\Phi_j$ will tend asymptotically towards zero as the distance from the discharging $z_j$ becomes large. That is to say, the net change in pressure which arises due to a junction has a finite value $P_j{}^*$ which is defined by:

$$P_j{}^* = \int_{z=0}^{z=\infty} \Phi_j(z)dz \tag{5}$$

Secondly, it is proposed that the developed flow pressure gradient $\Phi_d$ is constant for any given set of normalized velocities $U_a$ and $U_w$ That is to say:

$$P_d(z_j) = \Phi_d(U_a, U_w)\,z_j \tag{6}$$

Equations (5) and (6) form the basis of a two-phase flow hydraulic model which can be applied to drainage stacks. And shall now be validated using the data which has been gathered from three experimental annular flow loops ('test rigs').

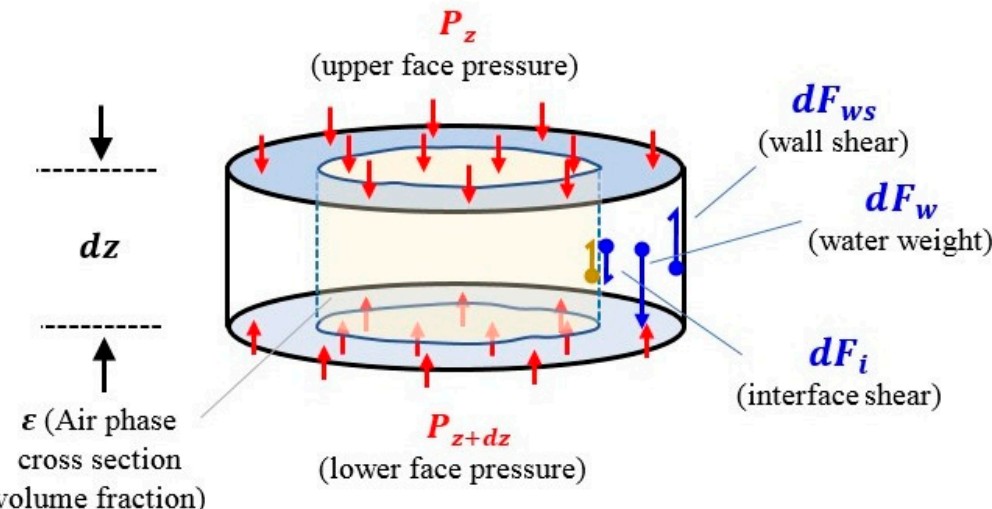

**Figure 3.** The forces and surface face pressures which act upon an element of a developing vertical-downwards annular flow.

### 3. Experimental Apparatus

A significant amount of test data has been collected for gravity-driven annular down-flow, using three different test rigs [28–30]. These rigs (labelled A, B and C in the discussion below) may be represented schematically by Figure 4a. Each rig comprises of an air inlet, a vertically downward-oriented test section hosting a water inlet, and an outlet. Each rig is instrumented with single-phase flowmeters, which record flowrates of air and water entering test sections, and with arrays of wall-mounted pressure transducers which record wall pressures within test sections.

A total of 107 tests have been conducted using Rigs 'A', 'B' and 'C' at the normalized velocities illustrated in Figure 2 (these velocities lie left of the annular flow transition boundary, and are representative of operating conditions expected to be encountered within high-rise buildings). In each test, wall pressures are collected by pressure transducers for steady-state flow are averaged to produce basis data which is reproduced in Sections 4 and 5. The three sets of basis data, comprised of 44 tests performed with Rig 'A', 27 tests with Rig 'B', and 36 with from Rig 'C', shall be referred to as the *basis datasets* in discussions below.

Notable physical differences exist between Rigs 'A', 'B' and 'C' which are summarized by Figure 4b. The test sections have different lengths, different diameters, and are made from different materials. The air flow is admitted either *actively* (i.e., under pressure) or *passively* (i.e., freely, without pressure). Rig 'A' is heavily corroded and discharges into a sewer, whilst Rig B and Rig C are made from smooth plastic materials and discharge into a collection tank. The locations at which water is injected into the test section vary between the test rigs, and the test sections are instrumented in different ways. In Rigs 'A' and 'C', pressure transducers are installed relatively far downstream of water inflow junctions, whereas in Rig 'B', pressure transducers are distributed through the test section downstream of the junctions.

The data generated from tests are now used to justify the proposals put forward in Section 2.1 above and to derive empirical correlations for the $\Phi_j$ and $\Phi_d$ pressure gradient components shown in Equation (4). It is to be noted that these correlations apply only to steady flow in straight pipe sections, bounded by the conditions shown in Figure 2. Nevertheless, these correlations are sufficiently robust to permit the analysis of drainage networks as is to be described in Section 6.

## Schematic Diagram

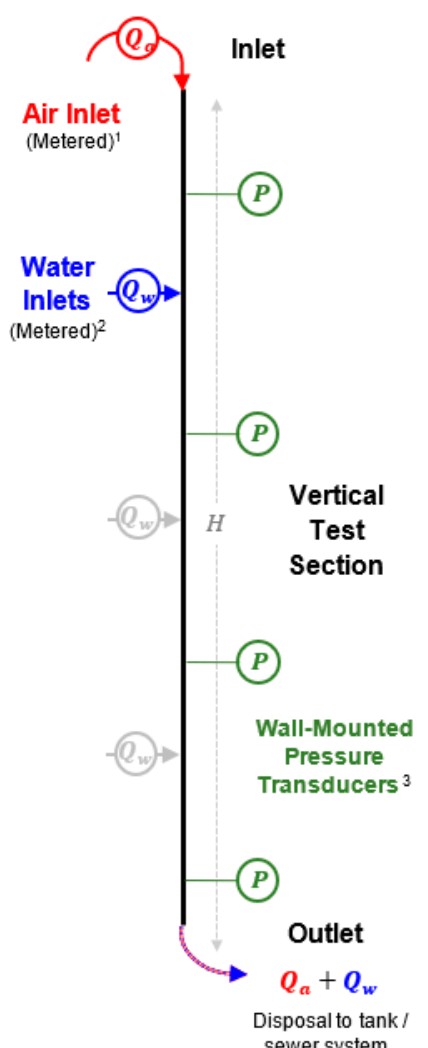

### Key Data

|  | **Rig 'A'** | **Rig 'B'** | **Rig 'C'** |
|---|---|---|---|
| Rig Description | Decommissi-oned drainage stack | Research drainage stack | Pressurized multiphase test loop |
| Stack Material | Corroded cast iron | Smooth plastic | Smooth plastic |
| Test Section ID | 152 mm[1] | 100 mm | 102 mm |
| Test Section elevation change | 44 m | 40 m | 6 m[2] |
| Elevations of water inflow junctions[3] | 14 m, 24 m, 34 m, 44 m [3] | 21 m, 24 m, 27 m, 30 m, 33 m, 36 m [3] | 0 m (at start of inlet 'vertical up' section [2]) |
| Method of Air Flow Control | Adjustable orifice at top of stack | None [4] | Valve located on outlet of compressed air supply tank |
| Water Disposal | Active sewer (via long horizontal drain) | Collection tank (via short horizontal pipe) | Collection tank (via short horizontal pipe) |
| Elevations of pressure transducers | 44 m, 34 m, 24 m, 14 m, 1.3 m [3] | 36 m, 33 m, 30 m, 27 m, 24 m, 21 m, 18 m, 15 m, 12 m, 9 m, 6 m, 3 m, 0 m [3] | 5 m, 1 m [5] |

Note 1: Actual ID is smaller than the 152mm stated, due to fact this rig has accumulated solid deposits during active service.
Note 2: This test section is located between a 6-metre 'vertical up' inlet section and a subsequent 6-metre 'vertical up' section connected by U-bends.
Note 3: Elevations are relative to the elevation of the test section outlet.
Note 4: Air flowrate is not actively controlled; it varies as a result of changes to water discharge location only.
Note 5: Elevations are relative to the base of the 6-metre 'vertical down' test section.

Note 1: In Rigs 'A' and 'C', the air inflow $Q_a$ is controllable via adjustable orifices.
Note 2: Rigs 'A' and 'B' host multiple water inlets along their test sections. One inlet is selected for use in each experimental test.
Note 3: The numbers of pressure transducers and their elevations are summarized in the table opposite.

(**a**)                 (**b**)

**Figure 4.** (**a**) Schematic diagram of gravity-driven annular downflow test rigs [28–30]. (**b**) Table summarizing key features.

## 4. 'Junction' Pressure Gradient Component

Figure 5a illustrates mean wall pressures which are obtained from the Rig 'B' basis dataset (comprised of the 24 sets of data obtained for conditions shown in Figure 2, at elevations shown in Figure 3). The data are displayed in the form of pressure change relative to the wall pressure above discharging junctions, P, versus normalized distances from junctions $z_j/D$. (The data obtained from the transducer located at the base of the stack, just above the 90° base elbow, are omitted).

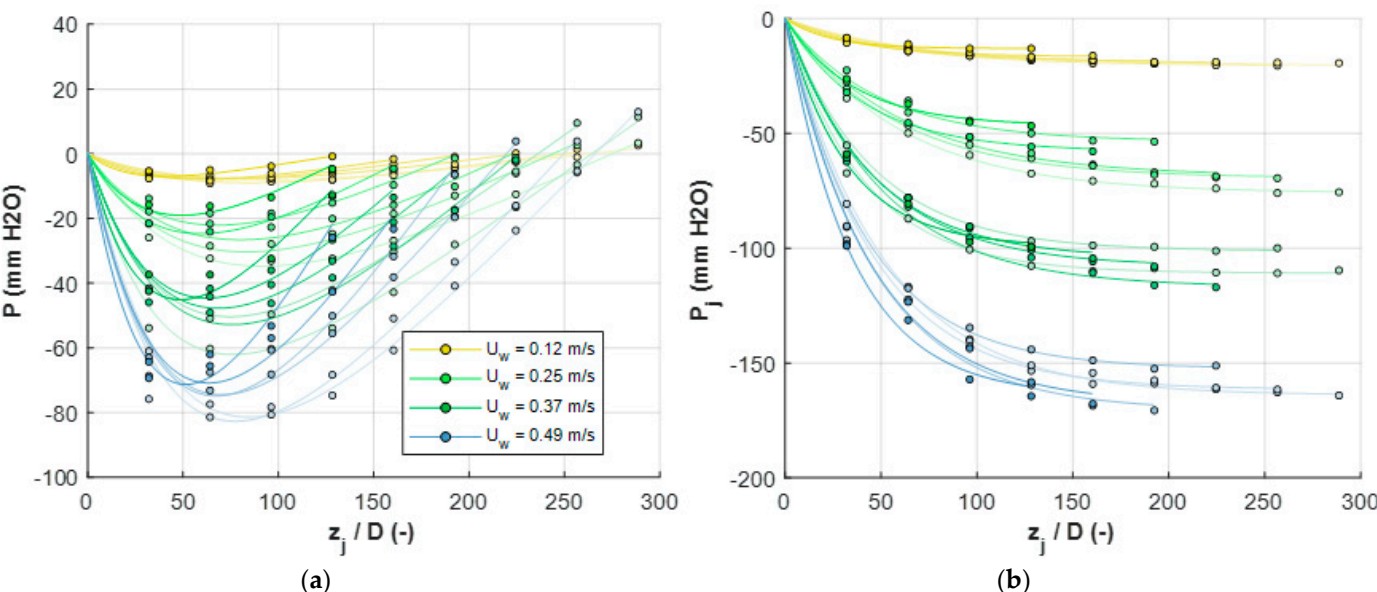

**Figure 5.** Analysis of Rig 'B' dataset. (**a**) Pressure loss P versus normalized distance $z_j/D$. (**b**) Junction pressure loss component $P_j$ versus normalized distance $z_j/D$.

The data shown in Figure 4a are interpreted more easily by superimposing trend lines which take the form:

$$P(z) = k_1\left(1 - e^{-k_2 z}\right) + \Phi_d z \tag{7}$$

where $\Phi_d$ is the developed flow pressure gradient component (defined in Equation (4) and evaluated using the techniques described in Section 5 below) and where $k_1$ and $k_2$ are empirical constants. The values of $k_1$ and $k_2$ are obtained from the data using the methods summarized in Table 1. These trendlines highlight that the pressure gradients increase monotonously with distance and tend towards limiting values as was proposed in Section 2.1 above.

**Table 1.** Derivation of the $\Phi_d$, $k_1$ and $k_2$ parameters within Equation (7).

| Parameter | Method of Calculation |
| --- | --- |
| $\Phi_d$ | Evaluated using correlation described in Section 5 below. |
| $k_1$ | Minimum value of the quantity $P_j$ (i.e., $P_j^*$ as shown within Figure 4b). |
| $k_2$ | Value which minimizes the quantity $\sum(ln(P - P_d - P(z)))^2$, using all data points *except* the data point used to calculate parameter $k_1$ as defined above. |

Figure 5b illustrates the residual pressure gradient component $P_j$ which is derived by evaluating the quantity $P - \Phi_d z$ and corresponding trendlines $P(z) - \Phi_d z$ which may be derived from Equation (7). These data suggest that the $P_j$ component tends asymptotically toward limiting values which are defined by Equation (5) and which are strongly dependent upon the water velocity $U_w$.

*Empirical Correlation*

An empirical correlation for the gradient $\Phi_j$ is developed as follows. The gradient $\Phi_j$ is assumed to be the product of a net junction loss parameter, $P_j^*$, and a distance-dependent decay parameter $\gamma$ according to the expression:

$$\Phi_j = P_j^*(U_a, U_w)\, \gamma\left(\frac{z_j}{D}, U_a, U_w\right) \tag{8}$$

These parameters $P_j^*$ and $\gamma$ are related to the parameters $k_1$ and $k_2$, used to fit trend lines as described in Table 1, as follows:

$$P_j^* = k_1\gamma = \frac{\Phi j}{P_j^*} = -k_2 e^{-k_2 z} \tag{9}$$

Figures 6 and 7 display values for $k_1$ and $k_2$ as a function of normalized velocities $U_w$ and $U_a$. The parameter $k_1$ is strongly dependent upon $U_w$ but weakly dependent upon $U_a$. The data suggest an empirical correlation for parameter $k_1$ may be developed which takes the form:

$$k_1 = \begin{cases} b_1 U_w + b_2 & \left(U_w > -\frac{b_2}{b_1}\right) \\ 0 & \left(U_w \le -\frac{b_2}{b_1}\right) \end{cases} \tag{10}$$

where the coefficient values $b_1 = -390$ mm $H_2O/ms^{-1}$ and $b_2 = 33$ mm $H_2O$ provide the fit to the data shown in Figure 6b. The constraint $k_1 = 0$ is imposed for $U_w < 0.085$ ms$^{-1}$, implying that there is no junction pressure gradient component for sufficiently low discharge water velocities.

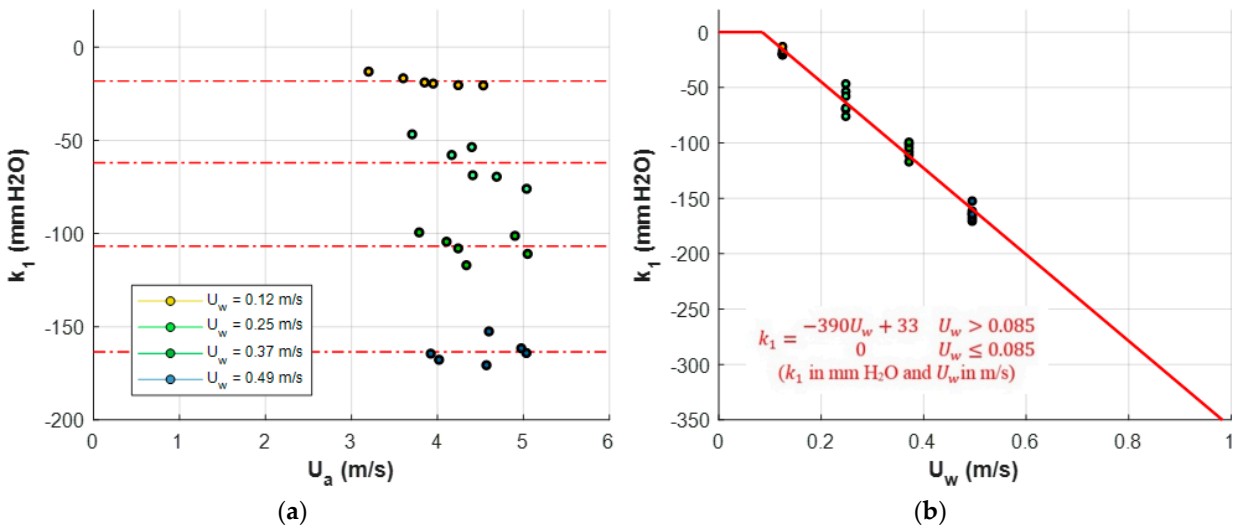

**Figure 6.** Empirical coefficient $k_1$ plotted as a function of (**a**) normalized air velocity $U_a$ (**b**) normalized water velocity $U_w$.

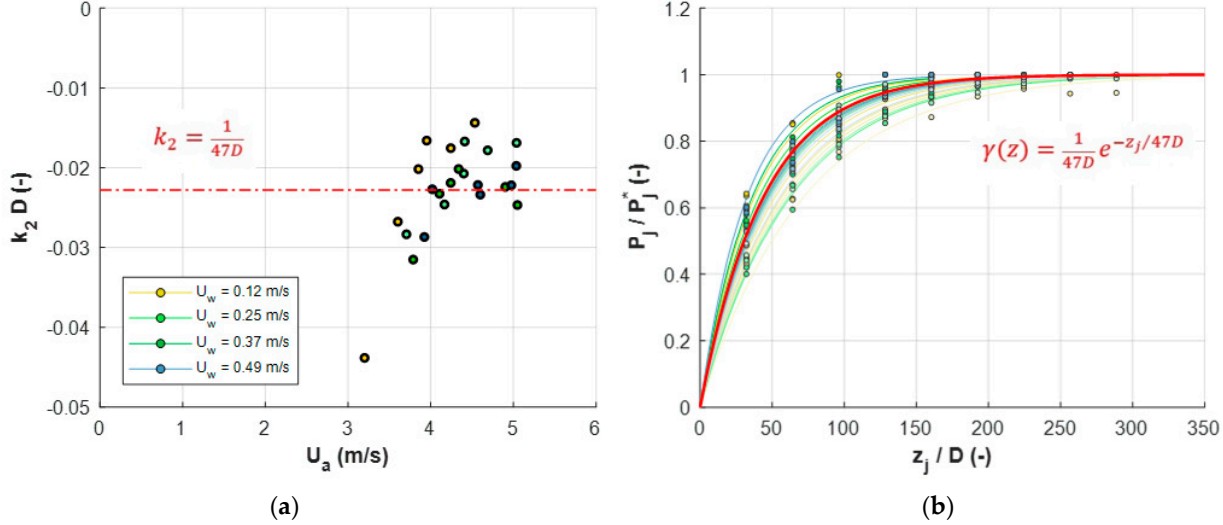

**Figure 7.** (**a**) Empirical product $k_2D$ plotted as a function of normalized velocity $U_a$ and (**b**) normalized junction contribution data, $P_j/P_j^*$, achieved with the mean coefficient value $k_2 = 1/47D$.

The parameter $k_2$ on the other hand appears to be strongly dependent on $U_a$ but rather weakly dependent on $U_w$. For simplicity, the average value:

$$k_2 = 1/47D \tag{11}$$

is assumed such that the single curve illustrated in Figure 7b defines the behavior of the junction pressure component $P_j$. This single parameter value provides a reasonable fit to all data.

Equations (8)–(11) are, of course, applicable for the one specific type of discharge junction installed within Rig 'B' spanning the range of normalized velocities shown within Figure 2. Comparison with Rig 'A' and Rig 'C' is not possible (Rigs 'A' and 'C' lack suitably located pressure transducers to enable such comparison to be made). However, experimental evidence suggests that junction losses are sensitive to the branch line diameter and angle of discharge from the branch line into the stack [31].

## 5. 'Developed Flow' Pressure Gradient Component

Figure 8 illustrates developed flow pressure gradient components $\Phi_d$ which may be derived from the Rig 'A', Rig 'B' and Rig 'C' datasets using data from transducers located downstream of junctions. The $\Phi_d$ values are plotted as a function of the air velocity $U_a$ and trend lines indicate constant values of velocity $U_w$. Each $\Phi_d$ value is derived from the expression:

$$\Phi_d \approx \frac{\overline{P}_Z - \overline{P}_Y}{\Delta z_{zy}} \tag{12}$$

where $\overline{P}_Z$ and $\overline{P}_Y$ are the pressure data obtained from the pressure transducers furthest downstream of junctions having separation distance $\Delta z_{zy}$. Note that the caveats shown in Table 2 are applied during calculation (which mean that strictly speaking, the data derived from Rigs 'A' and 'C' are estimates of $\Phi_d$).

The spread of data in Figure 8 reflects the significant physical differences between Rigs 'A', 'B' and 'C' that are summarized in Figure 4b. However, the $\Phi_d$ estimates are always positive for low $U_a$ values (i.e., pressure increases as vertical elevation decreases), and for these positive values, the trend lines indicate that:

1.  The derivative $\partial \Phi_d / \partial U_a$ is negative, i.e., pressure gradient $\Phi_d$ decreases with increasing air velocity.
2.  The derivative $\partial \Phi_d / \partial U_w$ is positive, i.e., pressure gradient $\Phi_d$ increase with increasing water velocity.
3.  The derivative $\partial^2 \Phi_d / \partial U_a \partial U_w$ is negative, i.e., rate of change of gradient $\partial \Phi_d / \partial U_a$ decreases with increasing water velocity.

All three sets of data display these trends, suggesting that they might be universal trends for vertical-downwards, gravity-driven annular flow.

**Table 2.** Notes which apply to calculation of the $\Phi_d$ components shown in Figure 8 using the Rig 'A' and Rig 'C' datasets.

| Rig | Notes |
|---|---|
| 'A' | The upstream pressure transducer Y is located *above* the discharge junction. However, test have been performed at low $U_w$ values, which would infer that the junction loss $P_j^*$ is zero (Figure 2, and Equation (5)). Hence, the junction contribution to $\Phi$ is expected to be negligible. |
| 'C' | Upstream pressure transducer Y is located a short distance downstream of a 180-degree 'U' bend. This bend may impact upon the flow development procedure. |

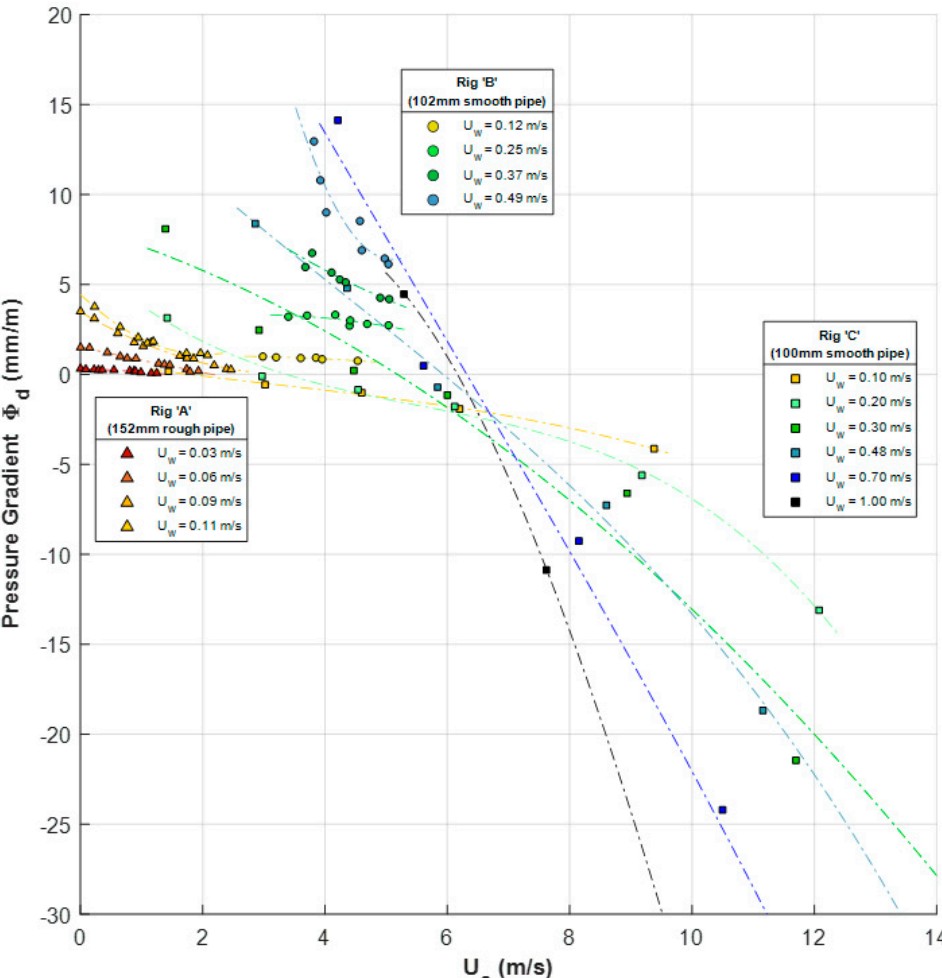

**Figure 8.** Developed flow pressure gradient components obtained from the Rig 'A', Rig 'B' and Rig 'C' datasets.

### 5.1. Empirical Correlation

Three empirical correlations for $\Phi_d$ are developed, on the basis that there are significant differences in flowrate conditions tested using Rigs 'A', 'B', and 'C'. Each empirical correlations is developed by assuming that $\Phi_d$ is a function of velocities $U_a$ and $U_w$ defined by the polynomial $g(U_a, U_w)$:

$$\Phi_d = g(U_a, U_w) = p_{00}U_a{}^i + p_{01}U_w{}^j + p_{20}U_a{}^{2i} + p_{11}U_a{}^iU_w{}^j + p_{02}U_w{}^{2i} + \\ p_{30}U_a{}^{3i} + p_{21}U_a{}^{2i}U_w{}^j + p_{12}U_a{}^iU_w{}^{2j} + p_{03}U_w{}^{3j} \tag{13}$$

where polynomial coefficients $p_{00}$ to $p_{03}$ are obtained by least-squares regression techniques, with exponent values $i = j = 1$ are applied as 'base case'. The appropriate coefficient values $g^A$, $g^B$ and $g^C$ for Rigs 'A', 'B' and 'C' are summarized in Table 3, while Figure 9a–c display the resultant functions $g^A$, $g^B$ and $g^C$. (These figures also show that by adjusting exponent values $i = j = 1.5$, extrapolation behavior the available range of data can be controlled).

Figure 9d illustrates *differences* between functions $g^A - g^C$ and $g^B - g^C$, for the limited regions of overlap between datasets shown in Figure 2 where this comparison is applicable. The difference $g^A - g^C$ is small, implying that Rigs 'A' and Rigs 'C' produce similar results. The difference $g^B - g^C$ is larger, indicating there is higher (positive) pressure gradient derived from Rig 'B' data. These differences reflect differences in annular film thickness, wall friction forces and interface friction forces between the test rigs, as well as possible pressure losses due to the stack geometry.

**Table 3.** Coefficient values for g($U_a$, $U_w$) correlation functions, Equation (13).

| Function | Exponents | | Polynomial Coefficients | | | | | | | | |
|---|---|---|---|---|---|---|---|---|---|---|---|
| | i | j | $p_{00}$ | $p_{10}$ | $p_{01}$ | $p_{20}$ | $p_{11}$ | $p_{02}$ | $p_{30}$ | $p_{21}$ | $p_{12}$ | $p_{03}$ |
| $g^A$ (Rig 'A') | 1 | 1 | 0.3207 | 0.7737 | −31.23 | −0.1885 | −25.17 | 1099 | −0.2629 | 15.89 | −156.6 | −4164 |
| $g^B$ (Rig 'B') | 1 | 1 | 27.22 | −27.15 | 116.6 | 8.55 | −65.98 | 188.2 | −0.8974 | 11.26 | −68.47 | 142.6 |
| $g^C$ (Rig 'C') [1] | 1 | 1 | −2.276 | −0.3256 | 30.43 | 0.316 | −14.18 | 87.49 | −0.0285 | 0.7723 | −3.461 | −41.24 |

[1] Based on data subject to restrictions $U_a < 10$ m/s and $\Phi_d > -10$ mm/m.

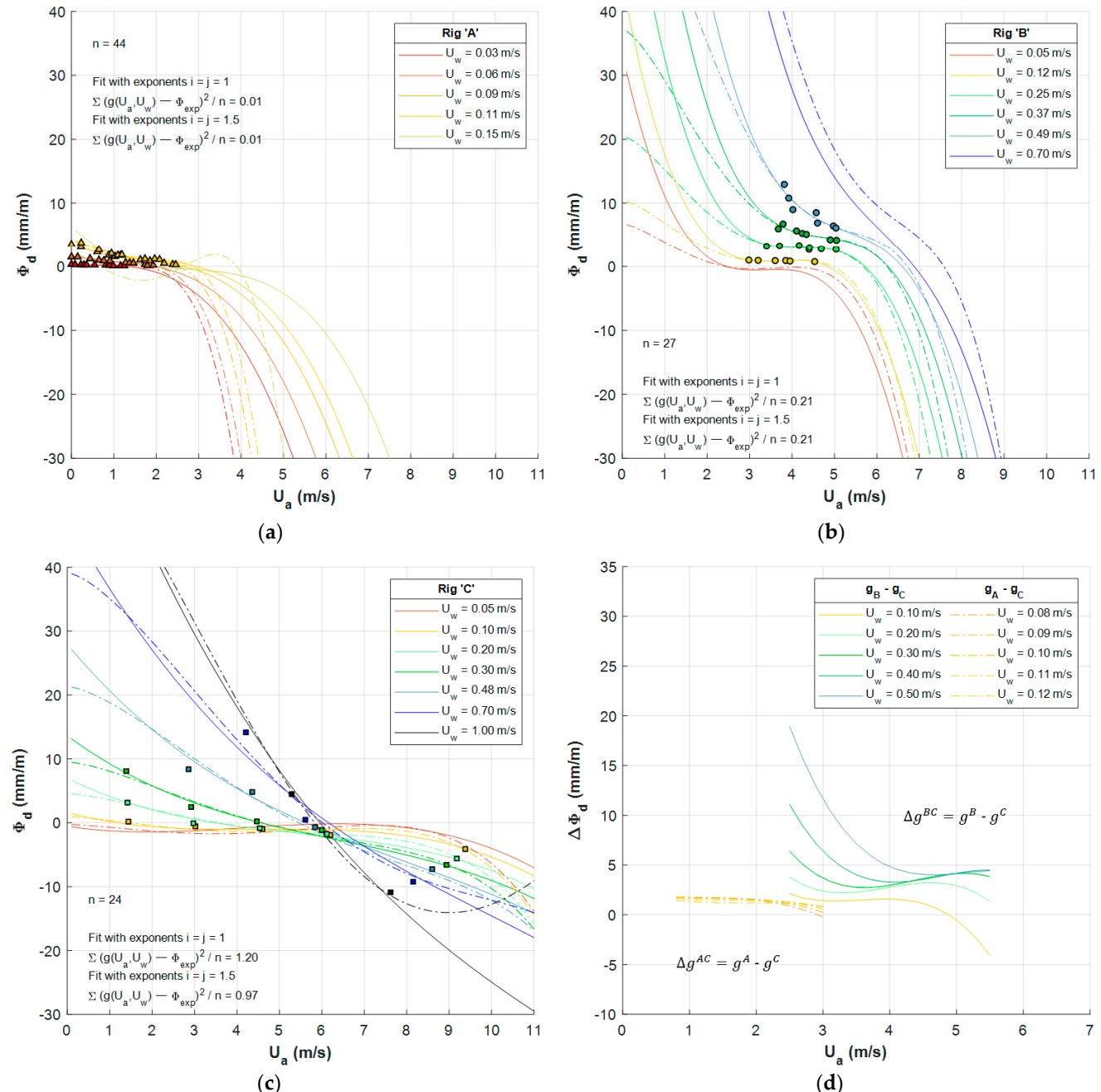

(**a**)  (**b**)  (**c**)  (**d**)

**Figure 9.** Empirical correlations for the $\Phi_d$ pressure gradient component defined by Equation (7). (**a**) Correlation for Rig 'A' dataset; (**b**) Correlation for Rig 'B' dataset; (**c**) Correlation for Rig 'C' dataset (**d**) Comparison between correlations, for regions of overlap indicated in Figure 2.

### 5.2. Limiting Air Velocity

A notable difference between the datasets plotted in Figure 8 is that the calculated $\Phi_d$ values are consistently positive for Rigs 'A' and 'B', whereas calculated $\Phi_d$ values eventually become *negative* as $U_a$ is increased for Rig 'C'. This difference arises as the air feeds in Rigs 'A' and 'B' is *passive*, whereas the air feed in Rig 'C' is *active* (that is to say, the air feed may be compressed such that it is drawn in at elevated pressure rather than atmospheric pressure). As Rig 'C' ejects air to atmosphere, this compression can support negative pressure gradients within the test section.

The air velocities at which the $\Phi_d$ values cross zero may be defined as *limiting velocities* ($U_a^L$). The data in Figure 6 suggest that the junction component $\Phi_j$ is consistently negative, and therefore, the $\Phi_d$ and $\Phi_j$ values will have opposing polarity on the condition that $U_a < U_a^L$. This opposing polarity ensures the downwards flow of air through the test sections. Moreover, this opposing polarity implies that the maximum air velocity which may be drawn through a *naturally ventilated* drainage stack *is* $U_a^L$, which applies regardless of stack height. This limit cannot be exceeded without performing work on the inflowing air stream.

Figure 10 illustrates 'limit velocity functions' $U_a^L(U_w)$ that are derived using the correlation coefficients listed in Table 3 (Rigs 'A' and 'B' these functions require data shown in Figure 9a,b to be extrapolated, to develop these functions for Rigs 'A' and 'B'. Therefore, the values shown in Figure 10 are estimate values for $U_a^L$ and they are displayed for a much smaller span of $U_w$ values than Rig 'C'). The functions take quadratic forms, reflecting the fact that the data in shown in Figure 9 have been nominally fitted using a cubic polynomial. Despite the physical differences between the three test rigs, the $U_a^L$ functions shown are very similar. The functions tend toward a plateau value of the order of 6 ms$^{-1}$ as $U_w$ is increased, suggesting that a maximum air velocity limit velocity of the order of 6 ms$^{-1}$ applies to all naturally ventilated vertical drainage systems. A similar tendency for the water velocity to plateau in this may be observed by analyzing data for large-diameter plunging dropshafts [32].

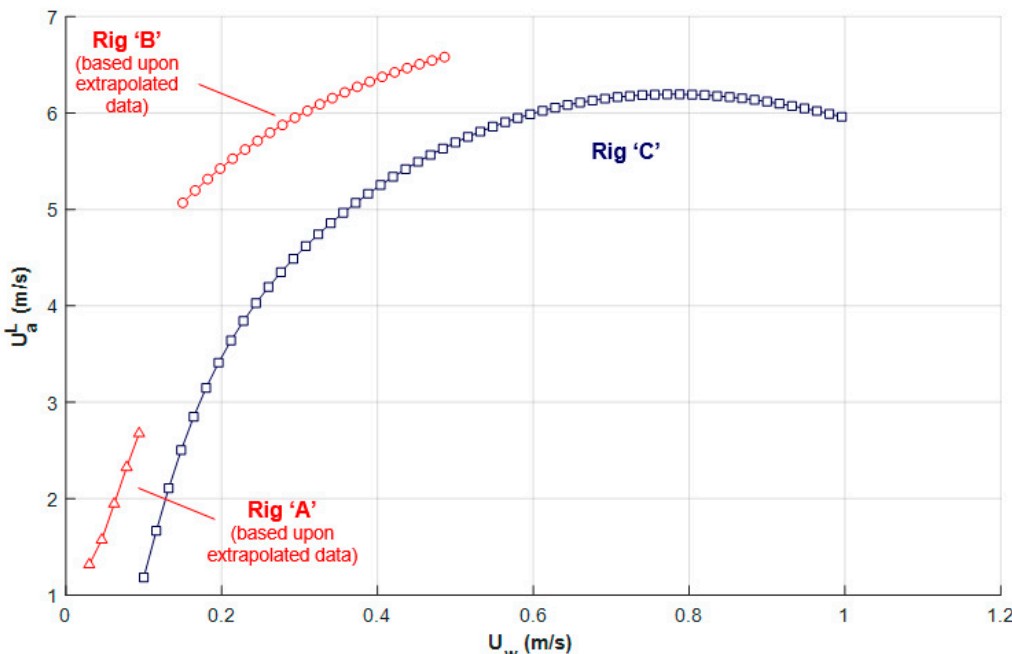

**Figure 10.** Limiting air velocities for Rigs 'A', 'B' and 'C', as derived from the empirical coefficients summarized in Table 3.

## 6. Network Analysis

Equations (8)–(11) and (13) can be employed to examine steady-state pressure profiles in a wastewater system network, such as the example system which is shown in Figure 11. This system is comprised of seven branches and seven nodes, with the wet stack accepting discharge water at nodes B and C (at velocities $U_w^B$ and $U_w^C$), and drawing in air from roof nodes A and B (at velocities $U_a^A$ and $U_a^E$. The boundary nodes A, D and E are open to atmosphere, while the vent line is permitted to exchange air with the wet stack via two crossover valves (XOVs). For design purposes the water velocities $U_w^B$ and $U_w^C$ may be treated as the system inputs while the air velocities $U_a^A$ and $U_a^E$ may be treated as unknowns.

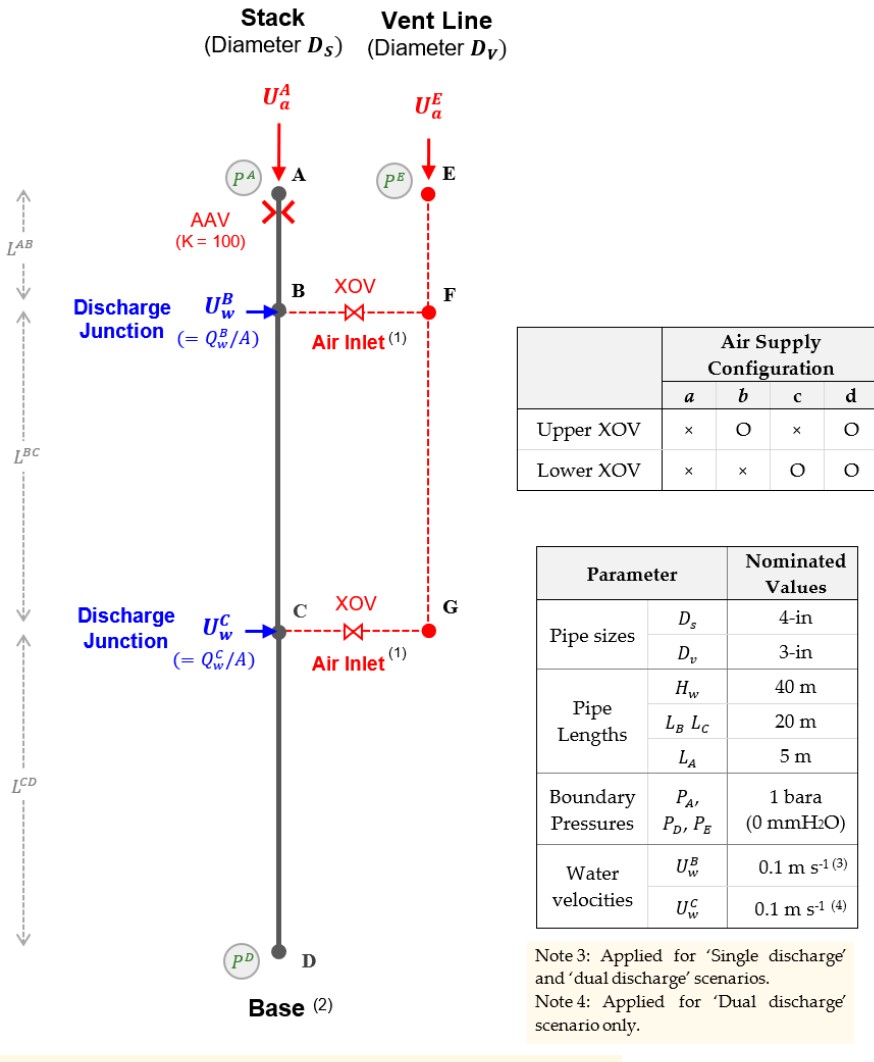

**Figure 11.** Drainage network schematic diagram. Nominated parameter values used to perform case studies described in Section 7 are summarized in the table.

The air velocities in branches, the hydraulic pressures at nodes, and the hydraulic pressure profiles between nodes shown in Figure 11 can be derived, provided that pressure gradients for flow in the dry branches ($\Phi_1(U_a)$) and the wet branches ($\Phi(U_a, U_W)$) are

supplied. If, for simplicity, the lengths of the crossover lines FB and CD are assumed to be zero, these parameters are obtained by solution of:

$$
\begin{pmatrix} 1 & 1 & 1 & 0 & 0 \\ 0 & 0 & 1 & 1 & 1 \end{pmatrix}
\begin{pmatrix}
P^{AB}\left(U_a^A\right) \\
P^{BC}\left(U_a^A + U_a^E, U_w^C\right) \\
P^{CD}\left(U_a^A + U_a^E, U_w^B + U_w^C\right) \\
P^{EF}\left(U_a^E\right) \\
P^{FB}\left(U_a^E\right)
\end{pmatrix}
= \begin{pmatrix} 0 \\ 0 \end{pmatrix}
\tag{14}
$$

where $P^{AB}$, $P^{BC}$, $P^{CD}$, $P^{EF}$ and $P^{FB}$ are the pressure losses across the branches. Equation (14) may alternatively be expressed as:

$$
\begin{pmatrix} L^{AB} & L^{BC} & L^{CD} & 0 & 0 \\ 0 & 0 & L^{CD} & L^{EF} & L^{FB} \end{pmatrix}
\begin{pmatrix}
\Phi_1\left(U_a^A\right) \\
\Phi\left(U_a^A + U_a^E, U_w^C\right) \\
\Phi\left(U_a^A + U_a^E, U_w^B + U_w^C\right) \\
\Phi_1\left(U_a^E\right) \\
\Phi_1\left(U_a^E\right)
\end{pmatrix}
= \begin{pmatrix} 0 \\ 0 \end{pmatrix}
\tag{15}
$$

where $L^{AB}$, $L^{BC}$, $L^{CD}$, $L^{EF}$ and $L^{FB}$ are the lengths of the branches shown in Figure 9. The single-phase (air) pressure gradients $\Phi_1$ may be derived from the classical expression:

$$
\Phi_1 = \frac{1}{2} f_D \rho_a U_a^2 / D
\tag{16}
$$

using an appropriate expression for the Darcy friction factor $f_D$. The two-phase pressure gradients $\Phi$ are evaluated Equations (8)–(11) and (13).

Two assumptions are now introduced in order to handle merging of fluids with the two-phase stream, as the discharge travels from branch BC to branch BD. The first assumption is that if the merging fluid is water, the junction pressure gradient component associated with this fluid is a function of the combined water flow velocity. Referring to node C within Figure 11, that is to say:

$$
\Phi_j^{CD} = f\left(U_a^A + U_a^E, U_w^B + U_w^C\right) \qquad \frac{\partial \Phi_j^{CD}}{\partial U_w^E} = 0
\tag{17}
$$

The second assumption is that if merging fluid is air, there is no junction pressure gradient component; i.e., there is no penalty associated with the air intake. Again, referring to node C in Figure 11, that is to say:

$$
\Phi_j^{CD} = 0 \qquad \frac{\partial \Phi_j^{CD}}{\partial U_a^E} = 0
\tag{18}
$$

Equations (17) and (18) close the hydraulic model, such that the drainage network shown by Figure 11 may be analyzed.

## 7. Preliminary Case Study

Solutions for a preliminary case study are now presented, based on the nominated parameter values shown in the table to the right of Figure 11 (i.e., a 40-m 4-in ID stack with a 3-in ID vent line, subject to 'single junction' and 'dual junction' discharges through nodes B and C). For this analysis the pressure gradient parameter $\Phi_d$ is evaluated using the $g^C$



function coefficients listed in Table 3 while the pressure drop across the air admittance value (AAV) is evaluated from the expression:

$$\Delta P = \frac{1}{2} K \rho_a U_a^2 \tag{19}$$

where the loss coefficient, K, has a nominated value of 100 as suggested by experimental testing [33].

Figure 12 illustrates air velocities in the network branches and pressure profiles though the stack for the two sets of simulation cases. The tables indicate that air flow through the network increases as the total water discharge rate increases and as the XOVs are opened. The air velocities approach, but do not exceed, the limiting values $U_a^L$ defined in Figure 7 when both XOVs are open (i.e., air supply configuration (d)). The graphs indicate that the suction pressures are most extreme when water is discharged through both nodes and when both XOVs are closed (minimum $-87$ mm $H_2O$ for the dual-discharge scenario with air supply configuration (a); with a notable contribution across the AAV). By opening the XOVs alleviates the suction pressures throughout the entire wet stack are alleviated.

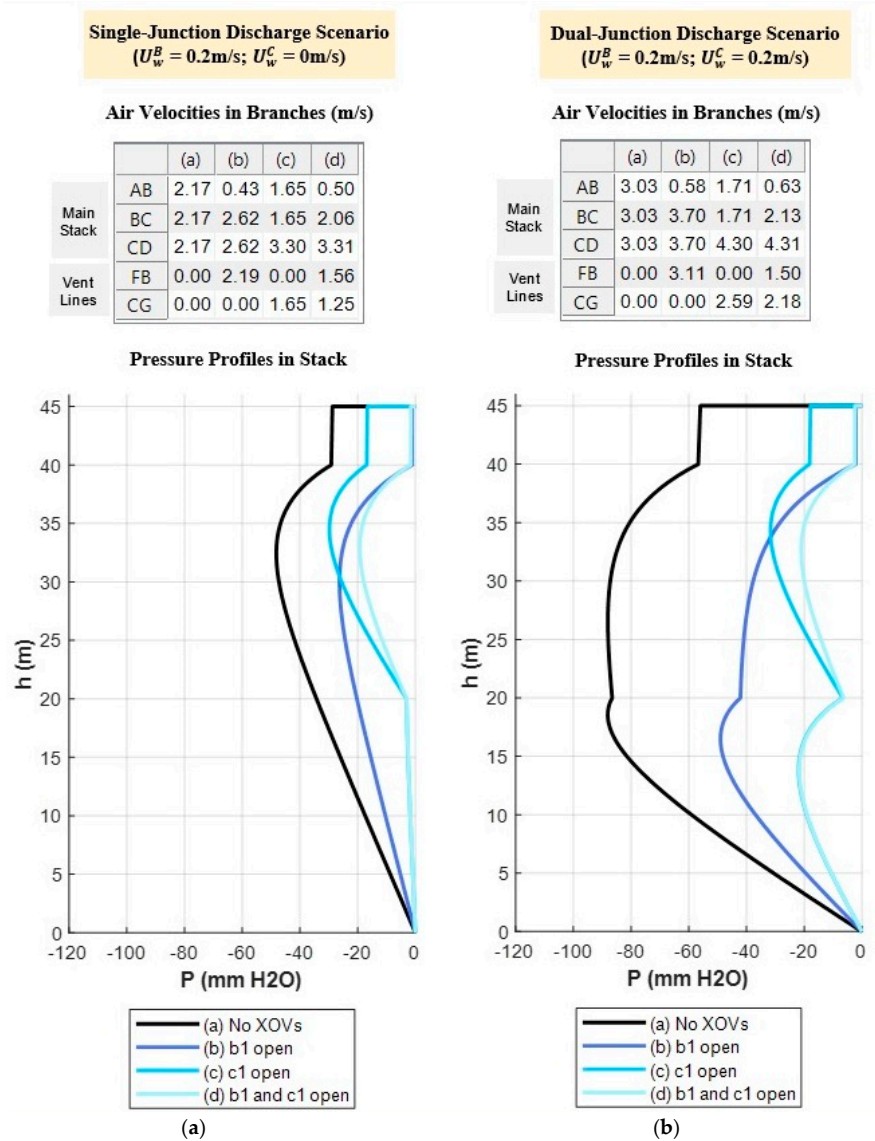

**Figure 12.** Preliminary case study results, for a $H_w = 40$ m stack, with stack diameter $D_s = 4$ inch and vent line diameter $D_v = 3$ inch. (**a**) Single junction discharge (**b**) Dual-junction discharge.

## 8. Summary

A model has been presented which allows the steady-state operating behaviour of vertical wastewater disposal system networks to be analysed. This model relates intake air flowrates and discharge water flowrates to the air velocities in branches and hydraulic pressure profiles. The model has been to examine behaviour of a 40-m 'medium-rise' vented-stack system hosting two junctions, crossvents and air admittance values, and to show how the air supply configuration and water discharge rates affect steady-state hydraulic pressure profiles. The model provides considerable insight which is not offered by existing design condes, and thus, has potential to be used as a tool for design of high-rise wastewater disposal systems.

**Author Contributions:** Conceptualization, C.S.; methodology, C.S.; software, C.S.; validation, M.G., D.K. and D.C.; formal analysis, C.S.; investigation, C.S.; resources, M.G., D.K. and D.C.; data curation, C.S.; writing—original draft preparation, C.S.; writing—review and editing, M.G., Y.X., D.K. and D.C.; visualization, C.S.; supervision, M.G., D.K. and D.C.; project administration, M.G., D.K. and D.C.; funding acquisition, M.G., D.K. and D.C. All authors have read and agreed to the published version of the manuscript.

**Funding:** This research was funded by Aliaxis S.A., and the APC was funded by Aliaxis S.A.

**Data Availability Statement:** The data obtained from [30] may be obtained by contacting the article author, Aliyu Aliyu (a.m.aliyu@hud.ac.uk). The remaining data may be obtained on request from correspondence author.

**Acknowledgments:** The authors gratefully acknowledge the assistance provided by Aliyu Aliyu of Huddersfield University and the support provided by Aliaxis Group which has enabled production of this article.

**Conflicts of Interest:** The authors declare no conflict of interest.

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
