# Peer review of "Steady-State Hydraulic Analysis of High-Rise Building Wastewater Drainage Networks: Modelling Basis"

_buildings, doi:10.3390/buildings11080344_

Round 1

Reviewer 1 Report

Reviewers' comments:

Manuscript ID: buildings-1265728

Title: Steady-State Hydraulic Analysis of High-Rise Building Wastewater Drainage Networks: Modelling Basis.

Manuscript Type: Article.

Reviewers' comments:

The manuscript describes the Steady-State Hydraulic Analysis of High-Rise Building Wastewater Drainage Networks: Modelling Basis. The manuscript needs a detailed editing. Some markings are made to just illustrate the extent of editing needed. A thorough revision addressing all the concerns is needed and if the authors are prepared to do that it can be considered for a review of the revised manuscript.

The authors need to consider the following comments

- Qualitative information’s are missing in abstract.

- Introduction is lacking of novelty statement. Please revise and add some recent papers in order to improve the introduction.

- The quality of figure 2 is too low.

- The Annular Flow Development section should be detailed.

- Figure 3 - is not clear make clear.

- Authors must but reference for each equation used.

- 5.1 Empirical Correlation – should be improve.

- Main findings should also be provided in conclusions.

- References: author should use order and there are recent references in 2020-2021 treating the same subject, you can use.

- Make all references in same format for volume number, page number and journal name, because it is difficult to searching and reading.

- Furthermore, they should add the graphical abstract, it is use full to readers.

- Several faults: are added or missing spaces between words: see PDF file (For example: Line numbers – 10, 44, 102, 103, 149, and 408, etc…)

Based on these, I advise the authors to rectify the above mentioned errors and we hope to re-evaluate the revised manuscript.

Reviewer 2 Report

Steady-State Hydraulic Analysis of High-Rise Building Wastewater Drainage Networks: Modelling Basis

In this work, a model and a methodology are presented which allow steady-state pressure profiles in high-rise wastewater drainage networks, subject to gravity driven annular flow, to be analysed.  The approach is justified by experimental observations and analysis. 

This paper is well written and can be recommended after some revisions.

  1. Some literatures may be added in line 26.
  2. such as [6], may be more clear?
  3. More available advances are advised to summarize.
  4. The theoretical model, empirical correlation, and experiment, as well case analysis are all presented, it is very helpfull to undertand your work.
  5. Φd is redefined in equation 7, please recheck all symbols.

Round 2

Reviewer 1 Report

Reviewers' comments:

The authors revised the manuscript according to the reviewers' comments.

So that I recommended this manuscript accept for publication in Buildings.